# Roles of RNA Methylations in Cancer Progression, Autophagy, and Anticancer Drug Resistance

**DOI:** 10.3390/ijms24044225

**Published:** 2023-02-20

**Authors:** Hyein Jo, Kyeonghee Shim, Dooil Jeoung

**Affiliations:** Department of Biochemistry, College of Natural Sciences, Kangwon National University, Chuncheon 24341, Republic of Korea

**Keywords:** autophagy, anticancer drug resistance, RNA methylation

## Abstract

RNA methylations play critical roles in RNA processes, including RNA splicing, nuclear export, nonsense-mediated RNA decay, and translation. Regulators of RNA methylations have been shown to be differentially expressed between tumor tissues/cancer cells and adjacent tissues/normal cells. N6-methyladenosine (m6A) is the most prevalent internal modification of RNAs in eukaryotes. m6A regulators include m6A writers, m6A demethylases, and m6A binding proteins. Since m6A regulators play important roles in regulating the expression of oncogenes and tumor suppressor genes, targeting m6A regulators can be a strategy for developing anticancer drugs. Anticancer drugs targeting m6A regulators are in clinical trials. m6A regulator-targeting drugs could enhance the anticancer effects of current chemotherapy drugs. This review summarizes the roles of m6A regulators in cancer initiation and progression, autophagy, and anticancer drug resistance. The review also discusses the relationship between autophagy and anticancer drug resistance, the effect of high levels of m6A on autophagy and the potential values of m6A regulators as diagnostic markers and anticancer therapeutic targets.

## 1. Regulators of RNA Methylation

RNA methylation acts as a critical regulator of gene expression. RNA methylations include N6-methyladenosine (m6A), N1-methyladenosine (m1A), 5-methylcytosine (m5C), 5-hydroxymethyl cytosine (hm5C), and N7-methylguanosine (m7G) [1,2,3]. N 6-methyladenosine (m6A) is the most prevalent epitranscriptomic internal modification of RNAs in eukaryotes [4,5,6,7]. m6A is enriched around stop codons, in 5′- and 3′-untranslated regions, and within long internal exons. m6A modifications occur in various RNAs, including messenger RNAs (mRNAs), ribosomal RNAs (rRNAs), transfer RNAs (tRNAs), circular RNAs (circRNAs), microRNAs (miRNAs), and long non-coding RNAs (lncRNAs). The effects of m6A modifications on non-coding RNAs (such as miRNA, lncRNA, and circRNA) include the modulation of pri-miRNA processing, lncRNA–protein, and lncRNA–RNA interaction, and circRNA transport [8]. m6A-mediated RNA methylations play a pivotal role in various cellular processes such as hematopoiesis, neurogenesis, cell differentiation, zygotic development, immune response, and tumorigenesis [9,10,11,12,13,14,15,16,17,18,19,20,21]. m6A regulators play crucial roles in various life processes by regulating RNA stability, mRNA splicing, translation or decay, and miRNA processing [17,18].

m6A modification is reversibly modulated through the installation of methyl transferases (writers), the removal of a methyl group by demethylases (erasers), and the recognition of m6A binding proteins (readers) [22,23]. m6A writers and associated proteins (m6A writer complex) include Vir-like m6A methyl transferase-associated protein (VIRMA) [24], methyl transferase-like protein 3 (METTL3) [25], METTL4 [26], MTTL5 [27], METTL14 [28], METTL 16 [29], Wilms’ tumor 1 associated protein (WTAP) [30,31], RNA-binding motif protein 15/15B (RBM15/15B) [32], casitas B-lineage lymphoma-transforming sequence-like protein 1 (CBLL1) [33], zinc finger CCCH-type containing 13 (ZC3H13), and zinc finger CCHC-type containing 4 (ZCCHC4) [5,34,35,36,37,38]. RBM15/15B, VIRMA, CBLL1, ZCCHC4, and ZC3H13 act as associated proteins of m6A writers. METTL3 and METTL14 induce RNA GAC, AAC, and GAC m6A methylations, respectively, and synergistically induce mRNA m6A modification [39,40]. METTL14 forms a stable heterodimer with METTL3 and plays a critical role in m6A deposition [41]. RBM15 and RBM15B are involved in X inactivation promoted by X-inactive specific transcript (XIST) [42].

WTAP forms a complex with METTL3 and METTL14 and is necessary for the efficient binding of a methyl transferase complex to RNA [43]. RBM15 [32,44], and RBM15B [45] recruit METTL3/METTL14/WTAP to target mRNA positions for m6A methylation. ZC3H13 is required for the nuclear localization of the ZC3H13-WTAP-VIRMAHAKAI complex and is, therefore, essential for m6A methylation. METTL16 deposits m6A into its specific messenger RNA targets [46]. METTL16 directly interacts with the eukaryotic initiation factors 3a and -b, as well as ribosomal RNA, thereby promoting the translation of over 4000 mRNA transcripts [46]. METTL16 catalyzes m6A methylation on S-adenosyl methionine (SAM) synthetase pre-mRNA and on A43 of U6 spliceosomal small nuclear RNA [47]. METTL16 regulates the expression of methionine adenosyl transferase 2A (MAT2A), which encodes SAM synthetase, thus controlling SAM homeostasis and mediating m6A methylation [48,49].

m6A demethylases (erasers) include fat mass and obesity-associated protein (FTO) and alkB homolog 5 (ALKBH5). m6A demethylases removes the m6A methylation group of RNA. FTO is involved in processing of micro RNAs (miRNAs) [50], as well as RNA stability [51] and metabolism [52]. ALKBH5 reduces m6A levels and is involved in mRNA export and RA metabolism [53]. 

m6A-binding proteins bind to the m6A methylation site. They include YT521-B homology (YTH) domain family proteins (YTHDFs), YTH domain-containing protein (YTHDCs) [54], insulin-like growth factor 2 mRNA-binding protein 3 (IGF2BPs) [55], eIF3 [46,56,57,58], and heterogeneous nuclear ribonucleoproteins (HNRNPs) [59]. IGF2BP2 enhances renal cell carcinoma tumorigenesis by stabilizing sphingosine-1-phosphate receptor 3 (S1PR3) mRNA [60]. HNRNPs mainly regulate alternative splicing or the processing of transcripts [61,62]. 

YTH domain-containing proteins comprise five functional genes: YTHDF1, YTHDF2, YTHDF3, YTHDC1, and YTHDC2. Lnc RNA Xist contains m6A modifications and is critical for X-inactivation. YTHDC1 binds to the m6A (UCG) hairpin of Xist and mediates X-inactivation [63]. Cytoplasmic m6A readers include YTHDF1/2/3, YTHDC2, and IGF2BP1/2/3. YTH domain-containing proteins first recognize the m6A modification of target RNAs, then direct the different complexes to regulate RNA splicing (YTHDC1/YTHDF2), nuclear export (YTHDF2), protein translation (YTHDF1), and RNA metabolism (YTHDF2) [64,65,66,67,68]. YTHDF2 binds to the m6A of target mRNA through its C-terminal YTD domain and accelerates the degradation of the target mRNA by enhancing the RNA translation of target mRNA in a cap-independent/m6A-dependent translation manner [69,70,71]. Table 1 and Figure 1 show the roles of m6A writers, demethylases, and m6A binding proteins in RNA metabolism. RNA metabolism includes miRNA processing, splicing, RNA decay, and many others, as indicated. Since RNA methylation regulates the expression of genes involved in various life processes, targeting RNA methylation can be used to produce anticancer drugs.

## 2. Role of METTL3 in Cancer

Since RNA methylation regulates the expression of various genes involved in cancer cell proliferation, METTL3 might regulate cancer initiation and progression. METTL3, an S-adenosyl methionine (SAM)-binding protein, transfers the methyl group from SAM to the adenine base in RNAs, generating S-adenosyl homocysteine (SAH) [72,73]. METTL3 recruits the YTHDC1 to induce the m6A methylation of DNA damage associated RNAs [74]. The METTL3-m6A-YTHDC1 axis induced DNA polymerase k to localize to DNA damage sites for nucleotide excision repair, and the recruitment of RAD51 and BRCA1 for homologous recombination–mediated repair in human sarcoma cells. [74]. METTL3 stabilized STEAP2 metallo reductase (STEAP2) mRNA and increased STEAP2 expression in an m6A-dependent manner in papillary thyroid cancer cells [75]. The downregulation of STEAP2 partially rescued the tumor-suppressive phenotype induced by METTL3 overexpression [75]. The METTL3-STEAP2 axis suppressed epithelial-mesenchymal transition (EMT) and the Hedgehog signaling pathway [75]. Thus, METTL3 can act as a tumor suppressor in papillary thyroid cancers. METTL3 was upregulated in human bladder cancer [76]. The overexpression of METTL3 promoted bladder cancer cell growth and invasion [76]. AF4/FMR2 family member 4 (AFF4), two key regulators of the NF-κB pathway (IKBKB and RELA), and MYC were identified as direct targets of METTL3-mediated m6A modification [76]. High levels of METTL3 could predict the poor prognosis of bladder cancer patients [77]. METTL3 in bladder cancer enhanced the maturation of pri-miR221/222, which decreased the expression of PTEN [77]. METTL3 promoted lung cancer cell growth, survival, and invasion by increasing the translation process of a set of target oncogenes, including epidermal growth factor receptor (EGFR) and the Hippo pathway effector tafazzin (TAZ) [78]. METTL3 was upregulated in cervical cancer (CC) tissue and cells, and high levels of METTL3 could predict poor prognosis of CC patients [79]. METTL3 recruited YTHDF1 to enhance HK2 stability and promote cell proliferation and the Warburg effect in CC cells [79]. Thus, METTL3 exerts oncogenic effects by promoting aerobic glycolysis. METTL3 and CDC25B were highly expressed in cervical cancer [80]. METTL3 induced CDC25B mRNA m6A modification in a YTHDF1-dependent manner and enhanced the tumorigenicity of CC cells in vivo [80]. Thus, METTL3 can serve as a target for developing anticancer drugs.

METTL3, YTHDF3, and YTHDF1 promoted yes-associated protein (YAP) translation by inducing the m6A modification of YAP in non-small cell lung cancer cells (NSCLCs) [81]. METTL3 increased the RNA levels of MALAT1 by inducing the m6A modification of MALAT1 [81]. MALAT1 promoted the invasion and metastasis of NSCLCs via YAP [81]. The downregulation of METTL3 inhibited the proliferation of NSCLCs (HCC827 and NCI-H1975), which could be restored by Fraser extracellular matrix complex subunit 1 (FRAS1) overexpression [82]. The downregulation of Fraser extracellular matrix complex subunit 1(FRAS1) or YTHDF1 inhibited the tumorigenic potential of NSCLCs [82]. METTL3 enhanced the stability of long non-coding RNA DLGAP1-AS2 in NSCLCs via m6A modification [83]. DLGAP1-AS2 downregulation suppressed the tumor growth of NSCLC cells [83]. DLGAP1-AS2 interacted with YTHDF1 to enhance the stability of c-Myc mRNA through DLGAP1-AS2/YTHDF1/m6A/c-Myc mRNA [83]. The downregulation of METTL3 in glioblastoma cells (GBM) decreased the m6A modification levels of serine- and arginine-rich splicing factors (SRSF), which led to the YTHDC1-dependent nonsense-mediated decay (NMD) of SRSF transcripts and decreased SRSF protein expression [84]. A decreased expression of SRSFs led to larger changes in alternative splicing isoform switches [82]. Importantly, the phenotypes mediated by METTL3 deficiency could be rescued by downregulating BCL-X or NCOR2 isoforms in GBM cells (GBM) [84]. Overall, these results establish a novel function of m6A in modulating NMD and uncover the mechanism by which METTL3 promotes GBM tumor growth and progression.

A high level of METTL3 could predict a poor prognosis in prostate cancer patients [85]. The downregulation of METTL3 decreased the expression of lymphoid enhancer binding factor 1 (LEF1) and inhibited Wnt signaling [85]. METTL3 was highly expressed in hepatocellular carcinoma (HCC), and the expression of ubiquitin-specific protease (USP7) was also increased [86]. METTL3 enhanced cancer cell proliferation by inducing the methylation of USP7 [86].

METTL3 was highly expressed in HCC tissues compared to adjacent tissues [87]. METTL3 depletion inhibited the M2 polarization of Kupffer cells (KCs), thereby suppressing the malignant phenotype of HCC cells [87]. METTL3 downregulation in KCs cells suppressed RNA binding protein 14 (RBM14) expression by decreasing m6A methylation [87]. The overexpression of RBM14 inhibited the anti-tumor effects of sh-METTL3 in vitro and in vivo [87].

High METTL3 expression could predict poor prognosis of patients with squamous cell carcinoma of the head and neck [88]. METTL3 downregulation inhibited the invasion, migration, and proliferation of oral squamous cell carcinoma (OSCC) cells by decreasing the m6A modification of protein arginine methyl transferase 5 (PRMT5) and programmed death-ligand 1 (PD-L1) [88]. METTL3 was upregulated in tissue samples, and high level of METTL3 could predict the poor prognosis of OSCC patients [89]. METTL3 targeted the 3′ UTR (near the stop codon) of the c-Myc transcript to install an m6A modification, thereby enhancing its stability [89].

The depletion of METTL3 in human hematopoietic stem/progenitor cells (HSPCs) promoted cell differentiation and inhibited cell proliferation [90]. METTL3 was highly expressed in acute myeloid leukemia (AML) cells compared to healthy HSPCs or other types of tumor cells [90]. m6A promoted the translation of c-MYC and BCL2 mRNAs in the human acute myeloid leukemia MOLM-13 cell line [90].

Adenosine deaminase acting on RNA 1 (ADAR1) and METTL3 were upregulated in breast cancer samples [91]. The loss of ADAR1 significantly inhibited breast cancer growth in vivo [91]. ADAR1 was shown to edit METTL3 mRNA and increases METTL3 protein, which further targeted Rho GTPase-activating protein 5 (ARHGAP5) in a YTHDF1-dependent manner [91].

The unfolded protein response (UPR) has been implicated in pancreatic ductal adenocarcinoma (PDAC) progression [92]. Nucleobindin 1 (NUCB1), a calcium binding protein, has been shown to control the UPR [92]. METTL3-promoted m6A modification of the NUCB1 5′UTR decreased the expression of NUCB1 in PDAC [92]. Low levels of NUCB1 could predict poor prognosis in patients with PDAC [92]. NUCB1 overexpression suppressed pancreatic cancer cell proliferation and showed additive effects with gemcitabine (GEM) in vitro and in vivo [92].

The m6A modification of PTEN mRNA by METTL3 decreased the expression of PTEN. LINC also decreased the expression of PTEN in chronic myelocytic leukemia cells [93]. The relative expression of LC3II, Beclin-1, ATG7, and ATG5 was decreased in cells treated with LINC00470, and down-regulated PTEN expression was observed in chemo-resistant cells [91]. Moreover, the downregulation of METTL3 also restored the normal levels of PTEN m6 A modification and LINC00470 expression in chemo-resistant cells [93]. Thus, METTL3 can acts as a tumor promoter in in chronic myelocytic leukemia cells. Taken together, these reports suggest that METTL3 acts as an oncogene or tumor suppressor in a context-dependent manner. Targeting METTL3 can be employed for developing anticancer drugs. Table 2 shows the roles and targets of METTL3 in cancer.

## 3. Role of METTL5 in Cancer

ZCCHC4 and METTL5 catalyze the m6A modification of 28S rRNA and 18S rRNA, respectively [105,106]. The 18S rRNA m6A methyl transferase complex METTL5-TRMT112 was upregulated in various cancer types and correlated with the poor prognosis of HCC patients [94]. The inhibition of METTL5-promoted 18S rRNA m6A modification decreased the translation of mRNAs involved in fatty acid metabolism and suppressed HCC progression [94]. METTL5 increased c-Myc translation to promote pancreatic cancer progression [95]. METTL5 expression was upregulated in breast cancer tissues and was necessary for the growth of breast cancer cell lines [107]. High METTL5 expression was associated with the poor prognosis of HCC patients [96]. The downregulation of METTL5 inhibited HCC cell proliferation, induced cell apoptosis, and decreased the expression of PD-L1, c-Myc, chaperonin-containing TCP1 subunit 2 (CCT2) and Chromobox 3 (CBX3) [96]. Table 1 shows the roles and targets of METTL5 in cancer.

## 4. Role of METTL14 in Cancer

METTL14 represses bladder cancer cell migration, invasion, and EMT by increasing the expression of USP38 [97]. METTL14 stabilized USP38 mRNA by inducing m6A modification and enhanced USP38 mRNA stability in a YTHDF2-dependent manner [97]. miR-3165 promoted bladder cancer (BCa) progression by targeting METTL14 expression [97]. METTL14-enhanced PTEN mRNA stability in a YTHDF1-dependent manner [98]. The upregulation of METTL14 inhibited clear cell renal cell carcinoma (ccRCC) cell proliferation by suppressing the activation of the phosphoinositide 3 kinase (PI3K)/AKT signaling pathway [98]. METTL14 promoted global genome repair (GGR) by regulating m6A mRNA methylation-mediated damage-specific DNA binding-protein 2 (DDB2) translation and suppressed ultraviolet B (UVB) radiation-induced skin tumorigenesis [99]. Skin-specific heterozygous METTL14 deletion increased UVB-induced skin tumorigenesis in mice. The downregulation of METTL14 induced changes in mRNA m6A enrichment and altered the mRNA expression of genes (e.g., ADAM19) [108]. m6A mRNA modification is critical for glioblastoma stem cell (GSC) self-renewal and tumorigenesis. The downregulation of METTL14 promoted human GSC growth, self-renewal, and tumorigenesis [108]. These reports suggest that METTL14 can act as a tumor suppressor in ccRCC and glioblastoma.

METTL14 was upregulated in choroidal melanoma compared to normal choroidal tissues [100]. METTL14 enhanced the invasion and migration of choroidal melanoma cells by activating Wnt/β-catenin signaling [100]. The downregulation of METTL14 inhibited cell proliferation and migration in lung cancer cell lines by increasing E-cadherin expression while suppressing N-cadherin and Twist expression [109].

METTL14 was highly expressed in normal hematopoietic stem/progenitor cells (HSPCs) and acute myeloid leukemia (AML) cells and was downregulated during myeloid differentiation [100]. METTL14 silencing promotes the terminal myeloid differentiation of normal HSPCs and AML cells and inhibits AML cell survival/proliferation. METTL14 was required for the development and maintenance of AML and the self-renewal of leukemia stem/initiation cells (LSCs/LICs) [101]. METTL14 exerted its oncogenic role by regulating its mRNA targets (e.g., MYB and MYC) through m6A modification [101]. These reports suggest that METTL14 can acts as an oncogene or tumor suppressor in a context dependent manner. Table 2 shows the roles and targets of METTL14 in cancer.

## 5. Role of METTL16 in Cancer

METTL16 was highly expressed in breast tumor tissues compared to adjacent tissues [102]. The downregulation of METTL16 decreased m6A methylation, suppressed the tumorigenic potential of breast cancer cells, induced ferroptosis, and enhanced the degradation of glutathione peroxidase 4 (GPX4) RNA in breast cancer cells [102]. METTL16 was upregulated in HCC, and a high expression of METTL16 could predict poor prognosis of HCC patients [103]. METTL16 induced the m6A modification of long non-coding RNA (RAB11B-AS1) and decreased the expression of the RAB11B-AS1 [103]. RAB11B-AS1 induced HCC apoptosis and suppressed HCC tumor growth [103]. Table 2 shows the roles and targets of METTL16 in cancer.

## 6. Roles of m6A-Binding Proteins and m6A Demethylases in Cancer

m6A regulators, such as m6A-binding proteins, can play roles in cancer initiation and progression. IGF2BP3 promoted the migration and invasion of triple negative breast cancer (TNBC) cells in an m6A modification-dependent manner [55]. IGF2BP3 bound to and destabilized the m6A-methylated mRNA of the extracellular matrix glycoprotein, slit guidance ligand 2 (SLIT2), to decrease the expression of SLIT and enhances the metastatic potential of TNBCs [55]. In doing so, IGF2BP3 activated canonical PI3K/AKT and MEK/ERK pathways [55].

High levels of YTHDF1 could predict the poor prognosis of patients with Merkel cell carcinoma (MCC) (58). The downregulation of YTHDF1 in MCCs inhibited the translation initiation factor eIF3 and proliferation [58]. YTHDF1 was upregulated in prostate cancer tissue, and high YTHDF1 expression could predict the poor prognosis of patients with prostate cancer [69]. YTHDF1 activated the polo-like kinase (PLK)/ PI3K/AKT axis to promote prostate cancer progression [69]. High levels of YTHDF1 could predict the poor prognosis of patients with HCC [104]. High levels of YTHDF1 were positively correlated with tumor size, and metastasis in breast cancer patients [110]. YTHDF1 promoted breast cancer metastasis by accelerating the translation of forkhead box M1 (FOXM1) [103]. The downregulation of YTHDF1 suppressed the proliferation and epithelial-mesenchymal transformation (EMT) and induced cell cycle arrest in breast cancer cells [111]. YTHDF1 was highly expressed in cisplatin-resistant colon cancer cells and increased the translation of glutaminase 1 (GLS1) to enhance colon cancer cell proliferation [110]. YTHDF1 enhanced the proliferation and cancer stem cell-like properties of glioblastoma cells [112]. YTHDF1 was the most upregulated in lung adenocarcinoma patients with KRAS/TP53-mutations and predicted the poor prognosis of patients with lung adenocarcinomas and enhanced the translation of cyclin B1 mRNA in an m6A-dependent manner [113]. Ataxia-telangiectasia mutated (ATM), a master in the DNA damage response, is modified by m6A epigenetic modification. The oncogenic potential of METTL3 and YTHDF1 results from the suppression of ATM expression via m6A modification [114].

A high expression of YTHDF1 was associated with more aggressive tumor progression and poor overall survival of patients with gastric cancers [115]. YTHDF1 increased the translation of a key Wnt receptor frizzled7 (FZD7) in an m6A-dependent manner, resulting in hyperactivation of the Wnt/β-catenin pathway and promoting gastric carcinogenesis [115]. YTHDF1 mediated the effect of the Wnt/β-catenin signaling pathway to enhance tumorigenicity and the stem cell-like activity of colorectal cancer (CRC) cells [116]. YTHDF1 enhanced the tumorigenic potential of HCCs by activating PI3K/Akt/mTOR signaling [117]. YTHDF1 is upregulated in intrahepatic cholangiocarcinoma (ICC), and high levels of YTHDF1 could predict the poor prognosis of ICC patients [118]. YTHDF1 increased the translation of EGFR mRNA via binding to m6 A sites in the 3′-UTR of the EGFR transcript, thus leading to the aberrant activity of downstream signal pathways that could promote tumor progression [118].

YTHDF2 promoted pancreatic cancer cell proliferation by enhancing EMT via yes associated protein (YAP) signaling and inhibiting TGF-β/SMAD signaling [119]. LncRNA STEAP3-AS1 interacted with the YTHDF2, and protected STEAP3 mRNA from m6A-mediated degradation, enabling the high expression of STEAP3 protein. Increased Fe^2+^ levels resulting from the high expression of STEAP3 inhibited glycogen synthase kinase-3 beta (GSK3β) activity by increasing the Ser9 phosphorylation of GSK3β, thereby activating Wnt signaling to support CRC progression [120].

YTHDF3 is highly expressed in ocular melanoma tissues, which is related to poor clinical prognosis [121]. YTHDF3 is required for the maintenance of cancer stem cell (CSC) properties and tumor initiation in ocular melanoma [122]. YTHDF3 promotes the translation of the catenin beta 1 (CTNNB1) and contributes to ocular melanoma propagation [122]. YTHDF3 is downregulated in CRC, whereas YTHDC1 is abundantly expressed in colon adenocarcinoma [121]. This implies that YTHDF3 can also act as a tumor suppressor in a context-dependent manner.

The downregulation of m6A demethylase ALKBH5 was correlated with increased m6A methylation in osteosarcoma cells/tissues compared to normal osteoblasts cells/tissues [123]. ALKBH5 overexpression suppressed osteosarcoma cell growth and induced apoptotic cell death via the m6A-based direct/indirect regulation of YAP [124]. The m6A-modified 5′UTR of pyruvate dehydrogenase kinase 4 (PDK4) positively regulated its mRNA stability via binding with the YTHDF1/eEF-2 complex and IGF2BP3, respectively. The demethylation of PDK4 m6A by the dm6ACRISPR system decreased the expression of PDK4 and glycolysis of cancer cells [124]. m6A/PDK4 promoted the tumor growth and progression of cervical and liver cancer [124].

m6A demethylase FTO is highly expressed and plays a critical oncogenic role in acute myeloid leukemia (AML) by targeting a cohort of critical transcripts, such as ankyrin repeat and SOCS box-containing 2 (ASB2) and retinoic acid receptor alpha (RARA) [125]. Table 3 shows the roles and targets of m6A binding proteins and demethylases in cancer.

## 7. Roles of RNA Methylases and m6A Binding Proteins in Anticancer Drug Resistance and Immune Checkpoint

An analysis of the cancer methylome revealed that global methylation patterns could regulate therapeutic resistance [126]. Alterations of the m6A modifications interferes with drug efficacy by modulating the expression of multidrug efflux transporters (e.g., ABCG2, ABCC9, and ABCC10), drug-metabolizing enzymes (e.g., CYP2C8), and drug targets (e.g., p53 R273H) [127]. For example, METTL3 confers imatinib resistance, and high levels of METTL3 could predict the poor prognosis of patients with gastrointestinal stromal tumors (GISTs) [128]. METTL3-increased the expression of MRP1 via binding with YTHDF1 and eEF-1 [128]. Alterations of the m6A modifications may protect cells from drug-mediated cell death by regulating DNA damage repair (e.g., p53, BRCA1, Pol κ, UBE2B, and ERCC1), downstream adaptive responses (e.g., critical regulators of apoptosis, autophagy, pro-survival signaling, and oncogenic bypass signaling), cell stemness, and the tumor microenvironment (e.g., ITGA6, ITGB3, and PD-1).

KIAA1429 promotes cell proliferation and enhances the resistance of gastric cancer cells to cisplatin by increasing FOXM1 expression via YTHDF1 [129].

METTL3-mediated m6A modification induces the chemo-resistance in acute myeloid leukemia (AML) cells by increasing the stability of integrin subunit alpha 4 (ITGA4) mRNA [130].

METTL3 installed m6A at the point-mutated codon 273 (G > A) of p53 pre-mRNA, and this m6A-RNA modification promoted a preferential pre-mRNA splicing. Consequently, the produced p53 R273H mutant protein induced multidrug resistance in colon cancer cells [131].

CircKRT17 and METTL3 were elevated in osimertinib-resistant lung adenocarcinoma (LUAD) tissues and cells [132]. The downregulation of METTL3 enhanced the sensitivity of LUAD cells to osimertinib by decreasing the expression of circKRT17 [128]. Additionally, 5-Fluorouracil (5-FU)-resistant rectal cancer cells showed increased levels of mRNA m6A and METTL3 [133]. The downregulation of METTL3 suppressed glycolysis and enhanced the sensitivity of CRC cells to 5-FU [134]. The inhibition of the METTL3 enhanced the sensitivity of pancreatic cancer cells to chemotherapy, especially to cisplatin, gemcitabine, and 5-FU, by inhibiting mitogen activated protein kinase (MAPK) signaling [134].

METTL14 is highly expressed in NSCLCs and enhances the resistance of NSCLCs to cisplatin by increasing m6A level of pri-miR-19a [28]. Long non-coding RNA (RHPN1-AS1) and METTL3 were overexpressed in ovarian cancer (OC) [135]. METTL3 could enhance the stability of RHPN1-AS1 by the m6A modification [135]. RHPN1-AS1 enhanced the proliferation and tumorigenic potential of OC by activating PI3K/AKT signaling in cisplatin-resistant OC cells [135]. METTL3/ METTL14 upregulation enhanced the resistance of oral squamous cell carcinoma (OSCC) to cisplatin by inhibiting the interleukin -17 (IL-17) signaling [136]. METTL3 and ETTL14 function as downstream targets of CEBPA divergent transcript (CEBPA-DT) to confer cisplatin resistance in oral cancer [136].

The CD133+ stem cells exhibited the upregulated expression of m6A mRNA and METTL3 [137]. METTTL3 enhanced the stability of poly [ADP-ribose] polymerase 1 (PARP1) by recruiting YTHDF1 to the 3′-UTR of PARP1 mRNA [137]. PARP1 could effectively repair DNA damage and confers resistance to oxaliplatin in gastric cancer [137].

High METTL14 expression was found in pancreatic cancer tissues compared to adjacent normal tissues [138]. The downregulation of METTL4 enhanced the sensitivity of pancreatic cancer cells to cisplatin [138]. YTHDF1 promoted the protein synthesis of glutaminase 1 (GLS1) to confer cisplatin resistance [110]. The recruitment of YTHDF1 to m6A-modified tripartite motif containing 29 (TRIM29) promoted TRIM29 translation in the cisplatin-resistant OC cells [139]. The downregulation of YTHDF1 suppressed the CSC-like characteristics of the cisplatin-resistant ovarian cancer cells [139]. Thus, YTHDF1 induces the CSC-like characteristics of cisplatin-resistant cancer cells by binding to m6A-modified TRIM 29. [139]. YTHDF1 facilitates S-phase entry, DNA replication, DNA damage repair, and accordingly, YTHDF1 downregulation enhanced the sensitivity of breast cancer cells to adriamycin and cisplatin, and Olaparib (a PARP inhibitor) by targeting E2F8 [140]. METTL3 increased the expression of PD-L1 via lncRNA MALAT1 in pancreatic cancer cells [141]. METTL3 increased PD-L1 mRNA levels in an m6A-IGF2BP3-dependent manner in breast cancer cells [142]. METTL3 downregulation induced the destabilization of PD-L1 mRNA in breast cancer cells [142]. The inhibition of METTL3 or IGF2BP3 enhanced anti-tumor immunity via PD-L1-mediated T cell activation [142]. JNK1 is necessary for the binding of c-Jun to the METTL3 promoter, which was shown to increase the expression of METTL3 and global RNA m6A levels in bladder cancer cells [143]. JNK1 increased the expression of PD-L1 by inducing the m6A modification of PD-L1 mRNA [143]. Since JNK1 increased the expression of METTL3 and PD-L1, JNK1 inhibitors can be used in combination with immune checkpoint inhibitors. FTO enhances the resistance of CRC cells to 5-FU by promoting apoptosis-inducing factor (SIVA-1) degradation via YTHDF2 [64]. Table 4 shows the roles and targets of m6A regulators in anticancer drug resistance.

## 8. The Regulatory Role of m6A Modifications in Autophagy

Autophagy maintains cellular homeostasis and enables cells to respond to stress by recycling their damaged cellular proteins, organelles, and other cellular components. Autophagy displays both pro- and anti-tumorigenic roles in a context-dependent manner. Epigenetic modifications, such as DNA methylations, histone modifications, and RNA methylations can regulate genes involved in autophagy [144]. Thus, m6A modifications can regulate the expression of autophagy related (ATG) genes. The post transcriptional regulation of ULK1 was altered by FTO, which led to the increased expression of ULK1 and induction of autophagy [145]. IGF2BP3 inhibited ferroptosis by binding to m6A methylated mRNA encoding anti-ferroptotic factors, including glutathione peroxidase 4 (GPX4), solute carrier family 3 member 2 (SLC3A2), and ferritin heavy chain H1 (FTH1), in lung adenocarcinoma cells [146]. Thus, it will be interesting to examine the relationship between autophagy and ferroptosis.

m6A plays a critical role in regulating macroautophagy/autophagy by targeting ATG5 and ATG7. The downregulation of FTO in OSCC cell lines decreased the expression of eIF4G1, enhanced autophagic flux, and inhibited tumorigenesis [147]. YTHDF2 captured eIF4G1 transcripts containing m6A and induced mRNA degradation, thereby promoting autophagy and reducing tumor occurrence [147]. m6A modification enhanced the stability of ZFAS1 [148]. Lnc RNA ZFAS1 is upregulated in tumor tissues and neural progenitor cells (NPC) [148]. ZFAS1 upregulated the expression of ATG10 and regulated autophagy by inhibiting the PI3K/Akt signaling pathway to promote the proliferation and migration of NPC cells [148]. High levels of YTHDF1 could predict the poor prognosis of patients with HCC [149]. YTHDF1 deficiency inhibited HCC autophagy, growth, and metastasis [149]. YTHDF1 enhanced the translation of autophagy-related genes ATG2A and ATG14 by binding to m6A-modified ATG2A and ATG14 mRNA, thus facilitating autophagy and the autophagy-related malignancy of HCC [149]. These reports suggest that high levels of m6A promote autophagy (Figure 2).

The downregulation of WTAP reduced the levels of m6A LKB1 mRNA, which led to the increased stability of LKB1 mRNA [150]. The downregulation of WTAP could upregulate the level of autophagy and inhibit HCC proliferation [150]. Thus, WTAP can promote cancer cell proliferation by inhibiting autophagy. The downregulation of FTO decreased the expression of ATG5 and ATG7, leading to the attenuation of autophagosome formation, thereby inhibiting autophagy in 3T3-L1 cells [151]. Upon FTO silencing, ATG5 and ATG7 transcripts with higher m6A levels were captured by YTHDF2, which resulted in mRNA degradation and reductions in protein expression, thus alleviating autophagy [151]. m6A modifications were increased in H/R-treated cardiomyocytes and ischemia/reperfusion (I/R)-treated mice hearts [152]. The downregulation of METTL3 enhanced autophagic flux and inhibited apoptosis in H/R-treated cardiomyocytes [152].

Autophagic flux enhanced by METTL3 deficiency was shown to be transcription factor EB (TFEB)-dependent [152]. METTL3 methylates TFEB, a master regulator of lysosomal biogenesis and autophagy genes, at two m6A residues in the 3′-UTR, which promotes the association of the RNA-binding protein HNRNPD with TFEB pre-mRNA and subsequently decreases the expression levels of TFEB [153]. TFEB induces ALKBH5 but inhibits METTL3. TFEB binds to the ALKBH5 promoter and activates its transcription. Thus, TFEB acts as a positive regulator of autophagy. Figure 2 shows that high levels of m6A inhibits autophagy.

## 9. The Regulatory Roles of m6A Writers in Autophagy

### 9.1. METTL14 Promotes Autophagy

Rapamycin-induced autophagy increased m6A RNA methylation via METTL14 in oral squamous cell carcinoma (OSCC) cells [154]. High METTL14 expression can inhibit OSCC growth [154]. METTL14 decreased the stability of eukaryotic translation initiation factor gamma 1 (eIF4G1) mRNA, suggesting that eIF4 acts as a negative regulator of autophagy [154]. Figure 3A shows that METTL14 mediates the effect of rapamycin on autophagy.

### 9.2. METTL3 Promotes Autophagy

Elevated m6A modifications induced by METTL3 enable YTHDF3 to promote autophagosome formation and lysosomal function upon nutrient deficiency [153]. YTHDF3 binding to the m6A modifications at the coding DNA sequence (CDS) and 3′ UTR around the stop codon of Foxo3 mRNA facilitated FOXO3 translation and induced autophagy [155]. Enterovirus-71 (EV-71) infection induced autophagy [156]. The downregulation of METTL3 prevented EV-71-induced cell death and suppressed EV-71-induced expression of Bax while rescuing Bcl-2 expression [156]. The downregulation of METTL3 inhibited the EV-71-induced expression of ATG5, ATG7, and LC3 II [156]. These reports suggest that METTL3 can promote autophagy (Figure 3A).

### 9.3. METTL1 Acts as Negative Regulator of Autophagy

tRNA m7G methyltransferase complex components METTL1 and WD repeat domain4 (WDR4) were highly expressed in esophageal squamous cell carcinoma (ESCC) tissues and associated with the poor prognosis of patients with ESCC [157]. Targeting METTL1 or WDR4 led to the decreased expression of m7G-modified tRNAs and reduced the translation of the oncogenes and negative regulators of autophagy in an m7G-related codon-dependent manner, which induced mammalian target of rapamycin complex C1 (MTORC1)-mediated autophagy via the dephosphorylation of Unc-51-like kinase 1 (ULK1) and cell death in ESCC [157]. Thus, METTL1 can inhibit autophagy to escape cell death. Figure 3A shows the effect of METTL1 on autophagy.

## 10. Cytoprotective Autophagy Promotes Anticancer Drug Resistance

Autophagy promotes anti-cancer drug resistance to facilitate cell survival, and the suppression of autophagy can enhance the sensitivity of cancer cells to anti-cancer drugs [158,159,160]. Cytoprotective autophagy can promote cell survival and adaptation. Aspartyl-tRNA synthetase 1 antisense 1 (DARS-AS1) was highly expressed in cervical cancer (CC) tissues compared to adjacent normal tissues [161]. DARS-AS1 facilitated DARS translation by recruiting METTL3 and METTL14, which bound with DARS mRNA 5′ UTR. Hypoxic exposure induced cytoprotective autophagy via the HIF1α/DARS-AS1/DARS axis [161]. METTL3 was highly expressed in lung adenocarcinoma tissues compared to paired normal tissues and conferred gefitinib resistance in NSCLC cells [162]. METTL3 promoted autophagy by increasing the expression of autophagy-related genes such as ATG5 and ATG7 [162]. β-elemene could overcome gefitinib resistance in NSCLC cells by suppressing cytoprotective autophagy exerted by METTL3 [162].

The activation of oncogenic tyrosine kinases (OncTKs) and receptor tyrosine kinases (RTKs) activation regulates autophagy via the PI3K/AKT/mTORC1 and RAS/Mitogen activated protein kinase (MAPK) signaling pathways. The targeted inhibition of tyrosine kinases (TKs) and RTKs have recently been considered for cancer therapy, but drug resistance continues to be a major limitation of tyrosine kinase inhibitors (TKIs). Manipulation of the autophagy pathway, along with the use of tyrosine kinase inhibitors (TKIs), may be a promising strategy to circumvent drug-resistance mechanisms.

## 11. The Roles of METTL3 in Anticancer Drug Resistance and Autophagy

### 11.1. METTL3 Promotes Autophagy to Induce Anticancer Drug Resistance

KIAA1429 recruited YTHDF1 to m6A modified FOXM1 and increased the expression of FOXM1, which induced cisplatin resistance (Figure 3B) [129]. This implies that high levels of m6A could induce anticancer drug resistance. PDGF ligands stimulated early growth response 1 (EGR1) transcription to induce METTL3 to promote glioblastoma stem cells (GSC) proliferation and self-renewal [163]. Targeting the PDGF-METTL3 axis inhibited mitophagy by decreasing the m6A modification of optineurin (OPTN) in GSCs [163]. Pharmacologic targeting of METTL3 augmented the anti-tumor efficacy of the PDGF receptor (PDGFR) and mitophagy inhibitors in vitro and in vivo [163]. These findings indicate the role of METTL3 in autophagy and anticancer drug resistance.

METTL3 confers resistance to gefitinib by promoting autophagy in NSCLCs (Figure 3B). The expression of METTL3 was highly increased in the cisplatin-resistant TCam-2 seminoma cell line [164]. METTL3 increased the expression of ATG5 by promoting m6A modification (Figure 3B) [164]. Long non-coding RNA ARHGAP5 promoted chemoresistance and its upregulation could predict the poor prognosis of gastric cancer patients [165]. METTL3 enhanced the stability of ARHGAP5-AS1 [165]. Autophagy was necessary for the increased the expression of ARHGAP5-AS1 in chemoresistant gastric cancer cells [165]. Figure 3B shows that METTL3 promotes autophagy to induce anticancer drug resistance.

### 11.2. METTL3 Inhibits Autophagy to Reverse Anticancer Drug Resistance

METTL3 is down-regulated in human sorafenib-resistant HCC [19]. The depletion of METTL3 under hypoxia promoted sorafenib resistance and angiogenesis in cultured HCC cells and activated autophagy-associated pathways [19]. METTL3 sensitized HCC cells to sorafenib by stabilizing forkhead box class O3 (FOXO3) in an m6A-dependent manner, thereby inhibiting the transcription of autophagy-related genes, including ATG3, ATG5, ATG7, ATG12, and ATG16L1 (Figure 3C) [19]. Thus, METTL3 inhibits autophagy to enhance anticancer drug sensitivity. The expression of FTO was increased in cisplatin-resistant (SGC-7901/DDP) gastric cancer cells [166]. The downregulation of FTO reversed cisplatin resistance in SGC-7901/DDP cells, which was attributed to the inhibition of (ULK1)-mediated autophagy (Figure 3C) [166]. Thus, high levels of m6A enhance the sensitivity of cancer cells to cisplatin by inhibiting autophagy.

## 12. RNA Methylome and Cancer Diagnosis

RNA methylation is a post-transcriptional level of regulation. More than 150 kinds of RNA modifications have been identified. These modifications are widely found in messenger RNA (mRNA), transfer RNA (tRNA), ribosomal RNA (rRNA), and various noncoding RNAs, including miRNAs and long non-coding RNAs. The number of methyl modifications to RNA is about seven times greater than for DNA. Since m6A regulators play important roles in cancer initiation and progression, they can be targets for the development of anticancer drugs. m6A regulators can also be employed as diagnostic and prognostic markers. With the development of high-throughput sequencing technology, the RNA methylome shows its clinical values.

Abnormal m6A modification levels caused by METTL3 have been identified as critical regulator in human cancers. High levels of METTL3 and CBLL1 could predict the poor prognosis of prostate cancer patients. The expression of METTL3, METTL14, WTAP, and CBLL1 was higher in prostate cancer cells compared to non-malignant prostate cells [167]. This suggests that m6A writers can be employed as prognostic markers.

High levels of METTL3 can predict poor overall survival (OS) and progression-free survival (PFS) in cancer patients, such as those with gastric cancer, ESOC, OSCC (*p* = 0.002), and other cancers [168]. Conversely, high levels of METTL14 were positively associated with better OS. High levels of KIAA1429 could predict poor OS (*p* = 0.001), and high levels of ALKBH5 were negatively associated with vascular invasion (*p* = 0.032) [168]. Thus, m6A regulators can predict the and PFS of various cancer patients.

Since m6A modification plays an important role in cancer, it is necessary to profile m6A modifications at a transcriptome-wide level. Transcriptomic and methylomic sequencing showed that METTL3-mediated m6A methylation modification was associated with the immune microenvironment and the effects of immunotherapy in GC patients [169]. Thus, alterations in the methylome by m6A regulators can predict the response to anticancer immunotherapy. m6A methylomic sequencing identified PD-L1 mRNA as a direct target of m6A modification whose levels were regulated by ALKBH5 [170]. ALKBH5 increased the expression of PD-L1 expression in intrahepatic cholangiocarcinoma (ICC) [170]. ALKBH5 deficiency increased m6A modification in the 3′UTR region of PD-L1 mRNA, thereby promoting its degradation [170]. High levels of ALKBH5 inhibited T cell-mediated antitumor immunity [170]. ALKBH5 expressing patients receiving anti-PD1 immunotherapy showed enhanced responses [170]. Thus, ALKBH5 could predict responses to anti-tumor immunotherapy. Methylated RNA immunoprecipitation sequencing (MeRIP-seq) and RNA sequencing (RNA-seq) identified two genes, Fc receptor-like 5 (FCRL5) and G protein-regulated inducer of neurite outgrowth 1 (GPRIN1), were associated with the prognosis and diagnosis of LUAD patients [171]. YTHDF1 was shown to regulate the expression of FCRL5 and GPRIN1 [171]. Genes with dysregulated m6A modifications were enriched in cancer signaling pathways [171]. Transcriptomic and m6A methylomic studies showed that adenocarcinoma of the esophagogastric junction (AEG) was correlated with the dysregulation of m6A RNA modifications [172]. MeRIP-seq and RNA-sequencing revealed four genes, WD- repeat domain 72 (WDR72), spectrin beta, non-erythrocytic 2 (SPTBN2), microrchidia 2 (MORC2), and prostate androgen-regulated mucin-like protein 1 (PARM1), which could predict the prognosis of patients with colorectal cancers [173]. CRC tissues showed differential m6A RNA modification patterns compared to adjacent normal tissues [173]. MeRIP-seq and RNA-seq data showed differential m6A RNA modifications between bladder cancer tissues and adjacent normal tissues [174]. Differentially methylated RNAs (mRNAs, lnc RNAs, circ RNAs) were enriched in caner signaling pathways [174].

Studies on the RNA methylome are in the initial stages. These studies will provide a better understanding of cancer initiation and progression. The RNA methylome could be the basis of cancer diagnosis and prognosis. Genes with aberrant m6A modifications can serve as targets for developing anticancer drugs.

## 13. Conclusions and Perspectives

Since METTLs regulate cancer initiation and progression, it will be interesting to examine the potential of METTLs as diagnostic markers of cancers. Therefore, the expression levels of METTLs in the tumor tissues and sera of cancer patients should be determined. The exosomes of anticancer drug resistant cells were shown to promote anticancer drug resistance and autophagic flux [159,175]. Thus, it is probable that exosomal miRNAs/cytokines can regulate the expression of METTLs to confer anticancer drug resistance or sensitivity.

Since METTLs are differentially expressed between tumor tissues and adjacent normal tissues, the transcription factors that bind to these METTLs should be examined. These transcription factors may serve as targets for developing anticancer drugs.

METTL3 can function both as an oncogene and tumor suppressor gene in a context-dependent manner. It is probable that METTL3 targets different genes in different cancers. Thus, it is necessary to identify molecular targets of METTL3 in various cancers. These targets will help us to better understand the mechanisms associated with autophagy and anticancer drug resistance.

Global levels of m6A RNA sequencing should be performed to determine the mRNA selectivity of m6A writers, demethylases, and m6A binding proteins. It is also necessary to identify the targets of m6A writers, erasers, and binding proteins in the same cancer cells and different cancer cells. Since tumors exhibit heterogeneity, it is probable that METTL3 targets different genes in the same cancer. It is also necessary to perform m6A RNA sequencing on a single-cell level in the same cancer.

Since m6A RNA modifications regulate RNA metabolism (RNA processing, RNA decay, RNA export, and translation), cancer initiation/progression, angiogenesis, cell proliferation, and cancer stemness, targeting m6A regulators could overcome resistance to anticancer drugs.

Since METTL3 is responsible for the increased expression of PD-L1 in various anticancer drug resistant cancer cells, the combination of a METTL3-targeting anticancer drug with an anti-PD-L1 antibody may overcome resistance to anti-PD-L1 blockage. It is necessary to identify small molecules that can bind to METTL3 and regulate the expression of METTL3. These small molecules can be developed as anticancer drugs. A phase I clinical trial (NCT05584111) of STC-15, an inhibitor of METTL3, is currently underway. This clinical trial will measure the efficacy and pharmacokinetics of STC-15 in 66 patients with advanced solid tumors. This trial will also determine the number of patients with adverse events, and the feasibility of using STC-15 as a cancer diagnostic marker. STC-15 oral capsules will be given once a day in 3-week cycles. Anticancer drugs targeting other m6A writers and regulators have not been developed. A complete understanding of the mechanisms of these molecules is necessary for developing anticancer drugs targeting m6A writers and regulators. Small molecules usually display systemic toxicity and low stability. It is known that exosomes act as delivery vehicle. Exosomes display low toxicity and high stability. Therefore, exosomes containing anticancer drugs can solve problems associated with free anticancer drugs.

MicroRNAs (miRNAs), small non-coding RNAs (18–24 nucleotides), regulate cancer initiation and progression [176], autophagy [177], and anti-cancer drug resistance [178]. miRNAs target multiple genes and play diverse roles. They can bind to the 3′UTR of target genes and inhibit translation or promote the degradation of target genes. Since METTLs regulate anticancer drug resistance, it is necessary to identify miRNAs that bind to the 3′UTR of these METTLs. Target scan analysis predicted that miR-21-5p, miR-590-5p, miR-493-5p, and miR-186-5p could bind to the 3′ UTR of METTL3 (personal observations). It will be interesting to examine the effects of these miRNAs on autophagy and anticancer drug resistance. These miRNAs can be developed as miR-mimics or miR-inhibitors to overcome anticancer drug resistance. Unlike siRNAs, only a few clinical trials have employed miR-mimics or miR-inhibitors.

Global RNA methylation patterns will provide the basis for the diagnosis and prognosis of cancers. Molecules targeting m6A regulators can be developed as anticancer drugs.

## Figures and Tables

**Figure 1 ijms-24-04225-f001:**
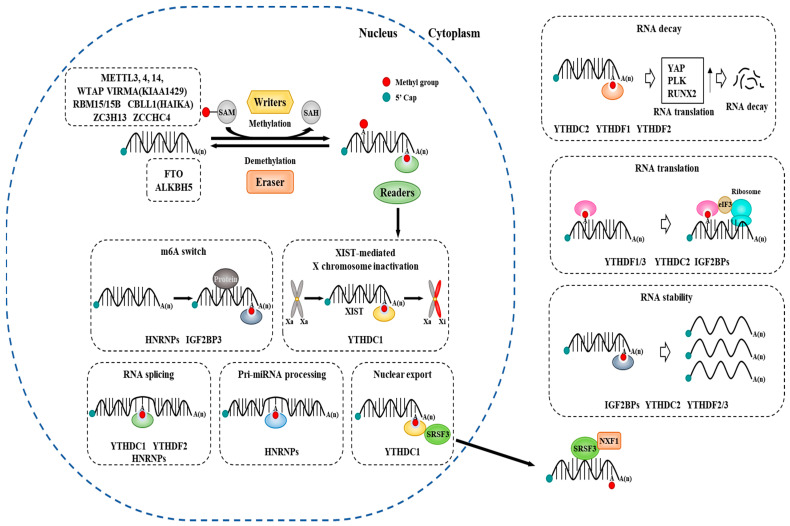
m6A RNA modification system. m6A methylation is catalyzed by writers, such as METTL3, METTL4, METTL5, METTL14, METTL16, WTAP, VIRMA, and others. Erasers, such as, FTO and ALKBH5, remove methyl groups on RNA. m6A binding proteins include YTHDF1-3, YTHDC1-2, HNRNPs, IGF2BP3, and others. m6A writers, erasers, and m6A binding proteins play critical roles in RNA metabolism, including RNA splicing, miRNA processing, m6A switch, RNA export, RNA decay, RNA stability, and RNA translation. Black arrows denote the direction of reaction.

**Figure 2 ijms-24-04225-f002:**
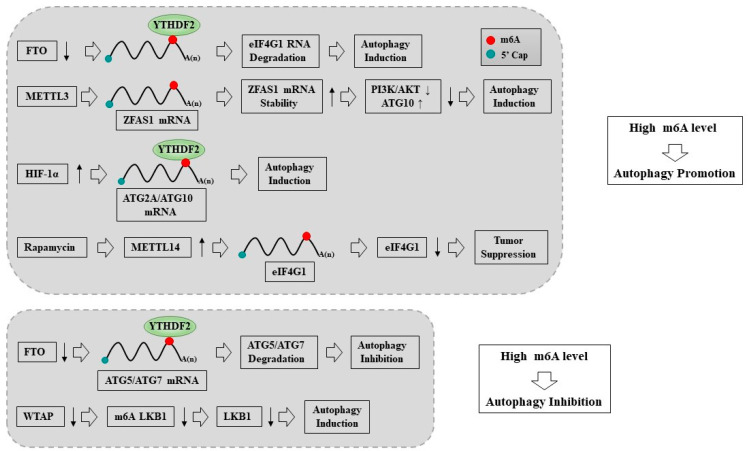
High m6A levels regulate autophagy in a context dependent manner. High levels of m6A promote autophagy (upper panel). High levels of m6A can be induced by METTL3, HIF-1α, METTL14, or downregulation of FTO (upper panel). High levels of m6A inhibit autophagy (lower panel). ↓ denotes decreased expression/activity. ↑ denotes positive regulation. Hollow arrows denote direction of reaction. ↓ denotes decreased expression/activity. ↑ denotes positive regulation. Hollow arrows denote direction of reaction. → denotes direction of reaction.

**Figure 3 ijms-24-04225-f003:**
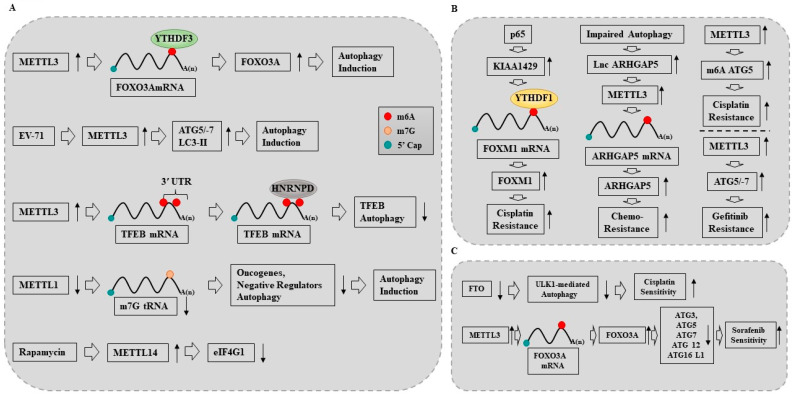
Roles of RNA methyl transferases in autophagy and anticancer drug resistance. (**A**) METTL3 promotes autophagy by increasing the expression of FOXO3A. METTL3 promotes autophagy by increasing the expression of ATG5/-7 and LC3-II. METTL3 inhibits autophagy by decreasing the expression of TFEB. METTL14 mediates the effect of rapamycin, an inducer of autophagy, by decreasing the expression of eIF4G1. Downregulation of METTL1 induces autophagy by decreasing m7G tRNA and oncogenes and negative regulators of autophagy. (**B**) METTL3 promotes or inhibits autophagy in a context dependent manner. (**C**) FTO enhances cisplatin sensitivity by promoting ULK1-mediated autophagy. METTL3 enhances anticancer drug resistance by promoting autophagy. KIAA1429 recruits YTHDF1 to increase the expression of FOXM1, which results in cisplatin resistance. ↓ denotes decreased expression/activity. ↑ denotes positive regulation. Hollow arrows denote direction of reaction. Hollow arrows denote direction of reaction.

**Table 1 ijms-24-04225-t001:** Roles of m6A regulators in RNA metabolism.

Type	Name	Functional Roles in m6A Regulation	Ref
m6A writers and regulators	VIRMA	Promotes m6A methylation of mRNAs in the 3′-UTR near the stop codons. A component of RNA methyl transferase complex	[24]
	METTL3	Induces RNA GAC, AAC methylation. A component of RNA methyl transferase complex	[25]
	METTL4	Induces N^6^-methylation of snRNA	[26]
	METTL5	Induces methylation of 18S rRNA, enhances translation	[27]
	METTL14	Induces GAC methylation. A component of RNA methyl transferase complex	[28]
	METTL16	Induces m6A methylation on S-adenosyl methionine (SAM) synthetase pre-mRNA	[29]
	WTAP	Enhances efficient binding of methyl transferase complex to RNA. A regulatory subunit of RNA methyl transferase complex	[30,31]
	RBM15/15B	Mediates X inactivation promoted by Xist. A component of m6A-METTL associated complex	[32]
	CBLL1	A component of m6A-METTL associated complex	[33]
	ZC3H13	Modulates RNA m6A methylation. A component of m6A-METTL associated complex	[34,35]
	ZCCHC4	Acts as rRNA N6-methyl transferase	[34,35]
m6A eraser	FTO	Involved in processing of miRNAs, RNA stability, RNA metabolism	[50,51,52]
	ALKBH5	mRNA export, methylation ↓	[53]
m6A binding proteins	IGF2BP1/2/3	Enhances tumorigenesis by stabilizing S1PR3	[60]
	HNRNPs	mRNA splicing	[61,62]
	YTHDC1	RNA splicing, X-inactivation	[63]
	YTHDF1	Enhances protein translation	[67]
	YTHDF2	RNA splicing, nuclear export, RNA degradation	[69]

↓ denotes decreased expression.

**Table 2 ijms-24-04225-t002:** Roles and targets of RNA methyl transferases in cancer.

Methylase	Target	Function	Mechanism	Cancer Type	Ref
METTL3	STEAP2	Tumor suppression	STEAP2 ↑, EMT ↑, Hedgehog signaling ↑	Papillary thyroid cancer	[75]
	NF-kB, MYC, AFF4	Tumor promotion	NF-kB ↑, MYC ↑, AFF4 ↑	Bladder cancer	[76]
	CDC25	Tumor promotion	m6A CDC25B ↑	Cervical cancer	[80]
	MALAT1	Tumor promotion	m6A MALAT1 ↑ YAP ↑	Non-small cell lung cancer	[81]
	DLGAP1-AS2	Tumor promotion	m6ADLGAP1-AS2 ↑, c-MYC ↑	Non-small cell lung cancer	[83]
	SRSF	Tumor promotion	m6ASRSF ↑	Glioblastoma	[84]
	LEF1, Wnt signaling	Tumor promotion	LEF1 ↑, Wnt signaling ↑	Prostate cancer	[85]
	C-Myc	Tumor promotion	C-Myc ↑, PRM5 ↑, PD-L1 ↑	Oral squamous cell carcinoma	[89]
	C-Myc, BCL2	Tumor promotion	C-Myc ↑, BCL2 ↑	AML	[90]
	NUCB1	Tumor promotion	NUCB1 ↓	Pancreatic ductal adenocarcinoma	[92]
METTL5	ACSL4	Tumor promotion	ACSL4 ↑	Hepatocellular carcinoma	[94]
	C-Myc	Tumor promotion	C-Myc ↑	Pancreatic cancer	[95]
	C-Myc, PD-L1	Tumor promotion	C-Myc ↑, PD-L1 ↑	Hepatocellular carcinoma	[96]
METTL14	USP38	Tumor suppression	USP38 ↑	Bladder cancer	[97]
	PTEN	Tumor suppression	PTEN ↑, PI3 kinase signaling ↓	Clear cell renal cell carcinoma	[98]
	GGR, DDB2	Tumor suppression	GGR ↑, DDB2 ↑	Skin tumor	[99]
	Wnt/β-catenin signaling	Tumor promotion	Wnt/β-catenin signaling ↑	Choroidal melanoma	[100]
	Myb, Myc	Tumor promotion	Myb ↑, Myc ↑	AML	[101]
METTL16	GPX4	Tumor promotion	GPX4 ↑	Breast cancer	[102]
	Cyclin D1	Tumor promotion	Cyclin D1 ↑	Gastric cancer	[103]
	RAB11B-AS1	Tumor promotion	RAB11B-AS1 ↓	Hepatocellular carcinoma	[104]

↓ denotes decreased expression. ↑ denotes positive regulation.

**Table 3 ijms-24-04225-t003:** Roles and targets of m6A-binding proteins and m6A demethylases in cancer.

m6A-Binding Protein/m6A Demethylase	Target	Function	Mechanism	Cancer Type	Ref
IGF2BP3	SLIT2	Tumor promotion	SLIT2 ↓, PI3K/AKT ↑, MAPK ↑	Breast cancer	[55]
IGF2BP2	S1PR3	Tumor promotion	S1PR3 ↑	Renal cell carcinoma	[60]
YTHDF1	eIF3	Tumor promotion	eIF3 ↑	Merkel cell carcinoma	[58]
YTHDF1	Polo-like kinase/PI3K/AKT	Tumor promotion	Polo-like kinase/PI3K/AKT ↑	Prostate cancer	[69]
YTHDF1	FOXM1	Tumor promotion	FOXM1 ↑	Breast cancer	[111]
YTHDF1	Cyclin B1	Tumor promotion	Cyclin B1 ↑	Lung adenocarcinoma	[113]
YTHDF1	FZD7	Tumor promotion	Wnt/β-catenin signaling ↑	Gastric cancer	[115]
YTHDF1	PI3K/Akt/mTOR	Tumor promotion	PI3K/Akt/mTOR signaling ↑	Hepatocellular carcinoma	[117]
YTHDF1	EGFR	Tumor promotion	EGFR ↑	Intrahepatic cholangiocarcinoma	[118]
ALKBH5	YAP, miR-181b-5p	Tumor suppression	YAP ↓, miR-181b-5p ↑	Osteosarcoma	[123]
FTO	ASB2 PARA	Tumor promotion	ASB2 ↓ PARA ↓	AML	[125]

↓ denotes decreased expression. ↑ denotes positive regulation.

**Table 4 ijms-24-04225-t004:** Roles and targets of m6A regulators in anticancer drug resistance.

Methylase/m6A Demethylase/m6A-Binding Protein	Function	Mechanism	Cancer Type	Ref
METTL14	Enhances cisplatin resistance	m6A level of pri-miR-19a ↑	Non-small cell lung cancer	[28]
	Enhances cisplatin resistance	AMPKα/ERK1/2/mTOR signaling ↑	Pancreatic cancer	[138]
FTO	Enhances 5-FU resistance	SIVA-1 ↓	Colorectal cancer	[64]
YTHDF1	Enhances cisplatin synthesis	GLS1 ↑	Colon cancer	[110]
	Enhances cisplatin resistance	TRIM29 ↑	Ovarian cancer	[139]
KIAA1429	Enhances cisplatin resistance	FOXM1 ↑	Gastric cancer	[129]
METTL3	Enhances cisplatin resistance	ITGA4 ↑	AML	[130]
	Enhances multidrug resistance	m6A installation at mutant p53	Colon cancer	[131]
	Enhances osimertinib resistance	CircKRT17 ↑	Lung cancer	[132]
	Enhances 5-FU resistance	Glycolysis ↑	Colorectal cancer	[133]
	Enhances cisplatin resistance	RHPN1-AS1 ↑, PI3K/AKT↑	Ovarian cancer	[135]
	Enhances oxaliplatin resistance	PARP1 ↑	Gastric cancer	[137]
	Enhances resistance to anti-PD-L1 blockage	PD-L1 ↑	Breast cancer	[142]
METTL3/ METTL14	Enhances cisplatin resistance	Functions as downstream targets of CEBPA-DT	Oral cancer	[136]

↑ denotes positive regulation. ↓ denotes decreased expression.

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
