# Peer review of "Roles of RNA Methylations in Cancer Progression, Autophagy, and Anticancer Drug Resistance"

_ijms, 2023, doi:10.3390/ijms24044225_

Round 1

Reviewer 1 Report

1. It is better to write m6A writers include methyl transferase complex types and their specifications briefly in a table.

2. There are similar studies and articles that are not listed in this article, and what is the difference between this article and them? They include:

doi: 10.1186/s12943-022-01680-z

doi: 10.1016/j.phrs.2021.105937

doi: 10.3389/fphar.2022.933332

doi: 10.3389/fonc.2020.01152

3. Mention clinical studies of target m6A regulators anticancer drugs in a separate section.

4. write futur directions in a section.

5. Write the limitations of using target m6A  regulator in cancer therapy.

Author Response

Dear Sir

Thanks for excellent suggestions.
I made changes to accommodate suggestions made by reviewers. I hope that changes I made are fine.

Sincerely yours

Jeoung Dooil 

Q1. It is better to write m6A writers include methyl transferase complex types and their specifications briefly in a table.

Ans. I made new table to better understand m6A methyl transferase complex and associated proteins. Please take look at new table 1.    

Q2. There are similar studies and articles that are not listed in this article, and what is the difference between this article and them? They include:

â‘ doi: 10.1186/s12943-022-01680-z

Liu, Z.; Zou, H.; Dang, Q.; Xu, H.; Liu, L.; Zhang, Y.; Lv, J.; Li, H.; Zhou, Z.; Han, X. Biological and pharmacological roles of m6A modifications in cancer drug resistance. Mol Cancer 2022, 21, 220. doi: 10.1186/s12943-022-01680-z.  ref.128

â‘¡doi: 10.1016/j.phrs.2021.105937

Yang, B.; Wang, J.Q.; Tan, Y.; Yuan, R.; Chen, Z.S.; Zou, C. RNA methylation and cancer treatment. Pharmacol Res 2021, 174:105937. doi: 10.1016/j.phrs.2021.105937. ref. 20

â‘¢doi: 10.3389/fphar.2022.933332

Song, N.; Cui, K.; Zhang, K.; Yang, J.; Liu, J.; Miao, Z.; Zhao, F.; Meng, H.; Chen, L.; Chen, C.; et al. The Role of m6A RNA Methylation in Cancer: Implication for Nature Products Anti-Cancer Research. Front Pharmacol 2022 13, 933332. doi: 10.3389/fphar.2022.933332. ref. 21

â‘£doi: 10.3389/fonc.2020.01152

Romero-Garcia, S.; Prado-Garcia, H.; Carlos-Reyes, A. Role of DNA Methylation in the Resistance to Therapy in Solid Tumors. Front Oncol 2020, 10, 1152, doi: 10.3389/fonc.2020.01152. ref.127

Ans. Thanks. I agree. I add the above references (20, 21, 127, 128). 

  1. Mention clinical studies of target m6A regulators anticancer drugs in a separate section.

Ans. I agree. I mention clinical trial of STC-15 (an inhibitor of METTL3). Please take look at lines 656-661. I tried to find clinical trials of m6A regulators. I have not seen any other clinical of m6A regulators.          

  1. write futur directions in a section.

Ans. I mention study on RNA methylome (Section 12: lines 574-625). I add new section on RNA methylome. Complete understanding roles of RNA methylome will provide clues to develop outstanding anticancer therapeutics, I also mention exosomes as anticancer drug carrying vehicle to enhance the efficacy of potential anticancer drug such as STC-15. Please take look at lines 665-667.       

  1. Write the limitations of using target m6A regulator in cancer therapy.

Ans. There are problems associated with using drugs targeting m6A regulators: 1) Inhibitors that target m6A readers, which mediate downstream effects of m6A-modified mRNAs, have not been discovered. The binding sites of a reader might be overlapping with other readers with opposed functions, which could have both beneficial and detrimental effects on cancer growth. 2) m6A writers have been known to act as both oncogene and tumor suppressor. 3)  the role and associated mechanism of the m6A modification in solid tumors still have not yet been fully understood. 4) Most inhibitors are at preclinical stages. Therefore, most inhibitors lack clinical values. 5) Most inhibitors lack selectivity. Therefore, these inhibitors can cause unwanted side effects. 6) m6A targeting inhibitors can impair anti-tumor immune response.   

* I send English certificate. In this revision, I sought help from English professionals.

Reviewer 2 Report

Jo et al. address the relevance of RNA methylations in cancer progression and anticancer drug resistance. In the current understanding, the role of modified nucleosides specifically RNA methylations in tumor hallmarks is relevant. However, the proposed review has significant scope to improve and make it better.

1.      The authors are encouraged to provide a concise introduction that should make better coherence from the Title to the Conclusion.

2.      The authors have attempted to include various forms of RNA methylation proteins, a discussion on the RNA methylome and diagnosis could be included.

3.      A discussion is needed on why autophagy in the context of RNA methylation and what about other forms of cancer cell death.

4.      Any preclinical and clinical trials on RNA methylome for diagnosis and therapeutics.

5.      Throughout the manuscript, the language, and flow of the paper should be improved. 

Author Response

Jo et al. address the relevance of RNA methylations in cancer progression and anticancer drug resistance. In the current understanding, the role of modified nucleosides specifically RNA methylations in tumor hallmarks is relevant. However, the proposed review has significant scope to improve and make it better.

Q1. The authors are encouraged to provide a concise introduction that should make better coherence from the Title to the Conclusion.

Ans. Thanks. I agree. I add references below to accommodate suggestions from other reviewer.   

â‘ Zheng, Z.H.; Zhang, G.L.; Jiang, R.F.; Hong, Y.Q.; Zhang, Q.Y.; He, J.P.; Liu, X.R.; Yang, Z.S.; Yang, L.; Jiang, X.; et al. METTL3 is essential for normal progesterone signaling during embryo implantation via m6A-mediated translation control of progesterone receptor. Proc Natl Acad Sci U S A 2023, 120, e2214684120. doi: 10.1073/pnas.2214684120. Ref.23

â‘¡You, Y.; Liu, J.; Zhang, L.; Li, X.; Sun, Z.; Dai, Z.; Ma, J.; Jiao, G.; Chen, Y. WTAP-mediated m6A modification modulates bone marrow mesenchymal stem cells differentiation potential and osteoporosis. Cell Death Dis 2023, 14, 33, doi: 10.1038/s41419-023-05565-x. Ref.29

In this revision, I try to remove unnecessary and repetitive sentences to make this manuscript more readable. I also fixed some wrong sentences. I also make new table (Table 1). I try to make Introduction as clear as possible.   

Q2. The authors have attempted to include various forms of RNA methylation proteins, a discussion on the RNA methylome and diagnosis could be included.

Ans. I add section on RNA methylome and cancer diagnosis (Section 12). Please take look at lines 574-625.    

A discussion is needed on why autophagy in the context of RNA methylation and what about other forms of cancer cell death.

Ans. The references below show the relationship between epigenetic modification and autophagy. Please take look at lines 432-436. m6A modifications can affect the expression of autophagy-related genes.

â‘ Shu, F.; Xiao, H.; Li, Q.N.; Ren, X.S.; Liu, Z.G.; Hu, B.W.; Wang, H.S.; Wang, H.; Jiang, G.M. Epigenetic and post-translational modifications in autophagy: biological functions and therapeutic targets. Signal Transduct Target Ther 2023, 8, 32, doi: 10.1038/s41392-022-01300-8. Ref 145

â‘¡Jin, S.; Zhang, X.; Miao, Y.; Liang, P.; Zhu, K.; She, Y.; Wu, Y.; Liu, D.A.; Huang, J.; Ren, J.; et al. m6A RNA modification controls autophagy through upregulating ULK1 protein abundance. Cell Res 2018, 28, 955-957. doi: 10.1038/s41422-018-0069-8. Ref.146

Q3. The references below show the relationship between m6A modifications and ferroptosis. Please take look at lines 436-441. m6A regulator can affect ferroptosis.    

â‘ Xu, X.; Cui, J.; Wang, H.; Ma, L.; Zhang, X.; Guo, W.; Xue, X.; Wang, Y.; Qiu, S.; Tian, X.; et al. IGF2BP3 is an essential N6-methyladenosine biotarget for suppressing ferroptosis in lung adenocarcinoma cells. Mater Today Bio 2022, ,17, 100503. doi: 10.1016/j.mtbio.2022.100503. ref.147

â‘¡Ye, F.; Wu, J.; Zhang, F. METTL16 epigenetically enhances GPX4 expression via m6A modification to promote breast cancer progression by inhibiting ferroptosis. Biochem Biophys Res Commun 2023, 638, 1-6. doi: 10.1016/j.bbrc.2022.10.065. ref.148

Q4.Any preclinical and clinical trials on RNA methylome for diagnosis and therapeutics.

Ans. I add section on RNA methylome and cancer diagnosis (section 12; lines 574-625). I also mention clinical trial of METTL3 inhibitor (lines 656-661). According to clinical trials.gov., I have not seen clinical trials of RNA methylome.

Q5. Throughout the manuscript, the language, and flow of the paper should be improved. 

Ans. Thanks. In this revision, I sought help from English professionals. I add English certificate.

Reviewer 3 Report

This review requires further meticulous self-examination. There are many inaccuracies in the text. The English requires professional refinement.

  1. The role of m6A mis-regulation in different cancers is contradictory, the author should specify the type of tumor when referring to m6A or some m6A regulators as tumor promoters or suppressors. Please check lines 116-117, 200, 240, and 441.
  2. The primary targets of METTL5, METTL16, and ALKBH3 are not polyA RNAs. In Figure 1, the author depicted all the regulators on polyA RNAs.
  3. The author should verify the use of full names for abbreviations. Please check lines 229, 359, 366, 370, 383, 427, and 610.
  4. Please review the accuracy of the sentence in lines 46-49. The listed genes do not belong to a single MTC.
  5. The information in lines 85-86 is redundant, as it is repeated in the first sentence of the same paragraph.
  6. There are inaccurate expressions in the text. Please review lines 104, 190-191, 195, 211-212, 223-224, and 267-269.
  7. In Section 6, FTO and ALKBH5 are m6A demethylases, not binding proteins. Please check the section and table titles.
  8. In Section 7, the author mistakenly classified FTO as a methylase/binding protein. Please review the section and table titles.
  9. Please revise all conflicting titles, such as the titles for Section 8 and Section 9.

Author Response

This review requires further meticulous self-examination. There are many inaccuracies in the text. The English requires professional refinement.

Q1. The role of m6A mis-regulation in different cancers is contradictory, the author should specify the type of tumor when referring to m6A or some m6A regulators as tumor promoters or suppressors. Please check lines 116-117, 200, 240, and 441.

Ans. - Thus, METTL3 can act as a tumor suppressor in papillary thyroid cancers (line 136).

-Thus, METTL3 can acts as a tumor promoter in in chronic myelocytic leukaemia cells (lines 218-219).

- These reports suggest that METTL14 can act as a tumor suppressor in ccRCC and glioblastoma (lines 254-255).

- I rather delete this sentence: This indicates that increased level of m6A modification enhances cancer cell proliferation by promoting autophagy.

Q2. The primary targets of METTL5, METTL16, and ALKBH3 are not polyA RNAs. In Figure 1, the author depicted all the regulators on polyA RNAs.

Ans. Thanks. I agree. I do not include METTL5, METTL16, and ALKBH3 in new figure (Figure 1).

Q3.The author should verify the use of full names for abbreviations. Please check lines 229, 359, 366, 370, 383, 427, and 610.

Ans. Thanks.

- [METTL14 (methyl transferase-like 14) promotes global genome repair] is changed into: METTL14 promoted global genome repair (GGR) through regulating  (line 246-247)

- [ METTL3 installs N6-methyladenosine (m6A) at the ] is changed into : METTL3 installed m6A at the point-mutated codon. (line 374)

- [ The CD133+ stem cells exhibited upregulated expression of N6-methyladenosine (m6A) mRNA and ] is changed into The CD133+ stem cells exhibited upregulated expression of m6A mRNA and METTL3. Lines 397-398.

- I delete the sentence: The N6-methyladenosine (m6A) RNA modification can regulate autophagy to modulate the initiation and progression of tumors.

Q4. Please review the accuracy of the sentence in lines 46-49. The listed genes do not belong to a single MTC.

Ans. Thanks. I agree. I change sentences into: m6A writers and associated proteins (m6A writer complex) include Vir-like m6A methyl transferase associated protein (VIRMA) [24], methyl transferase-like protein 3 (METTL3) [25], METTL4 [26], MTTL5 [ 27], METTL14 [28], METTL 16 [29], Wilms' tumor 1 associated protein (WTAP) [30, 31], RNA binding motif protein 15/15B (RBM15/15B) [32], Casitas B-lineage lymphoma-transforming sequence-like protein 1 (CBLL1) [33], zinc finger CCCH-type containing 13 (ZC3H13), and zinc finger CCHC-type containing 4 (ZCCHC4) [5, 34-38]. RBM15/15B, VIRMA, CBLL1, ZCCHC4, and ZC3H13 act as associated proteins of m6A writers. METTL3 and METTL14 induce RNA GAC, AAC, and GAC m6A methylation, respectively, and synergistically induce mRNA m6A modification [39, 40]. METTL14 forms a stable heterodimer with METTL3 and plays a critical role in m6A deposition [41]. RBM15 and RBM15B are involved in X inactivation promoted by X-inactive specific transcript (XIST) [42].

Please take look at lines 41-53.

Q5. The information in lines 85-86 is redundant, as it is repeated in the first sentence of the same paragraph.

Ans. Thanks. I change it. I delete this sentence: YT521-B homology (YTH) domain-containing proteins, including YTHDF1-3 and YTHDC1-2, confer m6A-dependent RNA binding activity.

Q6. There are inaccurate expressions in the text. Please review lines 104, 190-191, 195, 211-212, 223-224, and 267-269.

Ans.

  • I change the sentence into: Since RNA methylation regulates the expression of various genes involved in cancer cell proliferation, (please take look at lines 123-124)
  • I change the sentence into: METTL3-promoted m6A modification of the NUCB1 5'UTR decreased the expression of NUCB1 in PDAC [92]. Please take look at lines 207-208.
  • I change the sentence into : The m6A modification of PTEN mRNA by METTL3 decreased the expression of PTEN. LINC also decreased the expression of PTEN in chronic myelocytic leukaemia cells [93]. Please take look at lines 212-214.
  • I change the sentence into: The inhibition of METTL5-promoted 18S rRNA m6A modification decreased the translation of mRNAs involved in fatty acid metabolism and suppressed HCC progression [96]. Please take look at lines 229-231.
  • I change the sentence into: METTL14 stabilized USP38 mRNA by inducing m6A modification and enhanced USP38 mRNA stability in a YTHDF2-dependent manner [100]. Please take look at lines 240-242.
  • I change the sentence into: m6A regulators, such as m6A-binding proteins, can play roles in cancer initiation and progression. IGF2BP3 promoted the migration and invasion of triple negative breast cancer (TNBC) cells in an m6A modification-dependent manner [55]. Please take look at lines 284-286.  

    Q7. In Section 6, FTO and ALKBH5 are m6A demethylases, not binding proteins.

        Please check the section and table titles.

    Ans. Thanks. I change the section and table titles. Please take look at new table 3. Please take look at new manuscript.

    Q8. In Section 7, the author mistakenly classified FTO as a methylase/binding protein.

        Please review the section and table titles.

    Ans. Thanks. I check section and change table title. Please take look at new table 4.

    Q9. Please revise all conflicting titles, such as the titles for Section 8 and Section 9.

    Ans. I merge sections 8 and 9. New section title: The regulatory role of m6A modifications in                             Autophagy   

* I send English certificate. In this revision, I sought help from English professionals. 

Round 2

Reviewer 1 Report

All corrections have been made and the article is acceptable.

Author Response

Thank you for your kindness.

Reviewer 3 Report

Please revise all conflicting titles in Section 9 to Section 11.

Author Response

Dear Sir

I thank for excellent suggestions. I made changes. I hope that these changes are fine.

Sincerely yours

Jeoung Dooil 
